# Inhibition Effect of Ionic Liquid [Hmim]Cl on *Microcystis* Growth and Toxin Production

**DOI:** 10.3390/ijerph19148719

**Published:** 2022-07-18

**Authors:** Yang Liu, Yijie Zhang, Yousef Sultan, Peng Xiao, Li Yang, Hanyang Lu, Bangjun Zhang

**Affiliations:** 1College of Life Sciences, Henan Normal University, Xinxiang 453007, China; zhangyj2013@126.com (Y.Z.); sultanay18@hotmail.com (Y.S.); lhyang8168@163.com (H.L.); 2Department of Food Toxicology and Contaminants, National Research Centre, Dokki, Cairo 12622, Egypt; 3College of Life and Environmental Science, Wenzhou University, Wenzhou 325035, China; pxiao@wzu.edu.cn; 4Wuhan Imagination Science and Technology Development Co., Ltd., Wuhan 443000, China; bioyangli85@163.com

**Keywords:** [Hmim]Cl, toxicology, aquatic ecosystem, microorganism, photosynthesis

## Abstract

Ionic liquids (ILs) are known as “green solvents” and widely used in industrial applications. However, little research has been conducted on cyanobacteria. This study was conducted to investigate the toxicity of ionic liquids ([Hmim]Cl) on *Microcystis aeruginosa* PCC7806. The EC_50_ (72 h) of [Hmim]Cl on the growth of *Microcystis aeruginosa* PCC 7806 was 10.624 ± 0.221 mg L^−1^. The possible mechanism of toxicity of [Hmim]Cl against *M. aeruginosa* PCC 7806 was evaluated by measuring cell growth, photosynthetic pigment contents, chlorophyll fluorescence transients, cell ultrastructure, and transcription of the microcystin-producing gene (*mcyB*). The concentrations of chlorophyll *a* and carotenoids were significantly reduced in treated *M. aeruginosa* cultures. The results of chlorophyll fluorescence transients showed that [Hmim]Cl could destruct the electron-accepting side of the photosystem II of *M. aeruginosa* PCC 7806. Transmission electron microscopy demonstrated cell damage including changes in the structure of the cell wall and cell membrane, thylakoid destruction, and nucleoid disassembly. The transcription of the *mcyB* gene was also inhibited under [Hmim]Cl stress. In summary, this study provides new insights into the toxicity of [Hmim]Cl on cyanobactreia.

## 1. Introduction

Demand for green technologies has increased rapidly in recent years. Ionic liquids (ILs) have attracted great attention as promising organic solvents. ILs are low-temperature molten salts, which possess many unique physiochemical characteristics and excellent solvent properties (negligible vapor pressure, high thermal, chemical, and electrochemical stability) [1]. Therefore, ILs have been considered to be “green” substitutes for conventional volatile organic compounds (VOCs) used in synthetic and analytical processes [2]. Although ILs can reduce the potential risk of air pollution due to their negligible vapor pressures, they may pose ecological risks to aquatic ecosystems, due to their high solubility in water. In addition, many studies have found that ILs show poor biodegradability [3].

Algae are crucial primary producers for the food chain in aquatic ecosystems and play an important role in nutrient cycling. Algae have been used for hazard assessment of aquatic contaminants [4,5], because of their rapid response to environmental changes. Several studies have been done on the toxicity of ILs on aquatic organisms, such as green algae [6], diatoms [7], cladocerans [8], mussels [9], and fish [10]. However, little research has been conducted on cyanobacteria [11].

Global warming and eutrophication have enhanced the proliferation of cyanobacteria in freshwater bodies [12,13]. *Microcystis aeruginosa* (Kützing) Kützing 1846 is one of the most widespread and dominant cyanobacterial species [14,15]. *M. aeruginosa* also has the capability of producing secondary metabolites such as microcystin, the synthesis of which is sensitive to the surrounding environment [16,17]. Thus, the response of *M. aeruginosa* to IL stress can be used for assessing the impact of ILs on aquatic ecosystems. Oxygenic photosynthesis of cyanobacteria is easily affected by hazardous substances such as ILs [18]. The chlorophyll fluorescence measurement is an important index to evaluate the performance of photosynthesis processes and is usually measured via the redox states of photosystem II (PSII) and the electron-donating and -accepting sides of the active reaction centers. Krause et al. [19] and Strasser and Strasser [20] established the JIP-test, which conducts a kinetic analysis of the light-induced chlorophyll fluorescence rise curve based on the theory of biofilm energy flow. The JIP-test reflects the physiological state of cyanobacterial PSII by detecting the effects of different factors on its structure and function [18,21,22].

The compound 1-hexyl-3-methylimidazolium chloride, which is abbreviated as [Hmim]Cl, is one of the widely used ILs [23]. Until now, there has been no available study concerning the effects of [Hmim]Cl on cyanobacteria. The current study aims to illustrate the possible toxicological mechanism of [Hmim]Cl on cyanobacteria from the view of cell growth, photosynthesis, and gene expression.

## 2. Materials and Methods

The *M. aeruginosa* PCC 7806 was provided by the Institute of Hydrobiology, Chinese Academy of Sciences, and was cultured in a BG11 liquid medium [24]. The culture was maintained at a light–dark cycle of 12 h:12 h with a light intensity of 3000 lx under a controlled temperature (28 °C). The [Hmim]Cl (99% purity) was purchased from the Chengjie Chemical Company (Shanghai, China).

### 2.1. Measurement of Cell Growth and Photosynthetic Pigments Content

The [Hmim]Cl was prepared and added into the *M. aeruginosa* PCC 7806 culture with a cell density of 10^4^ cells mL^−1^. The final concentrations of [Hmim]Cl were set at 5, 10, 20, and 30 mg L^−1^. All experiments were performed in triplicate. The cell density and pigment contents in the control and treated cultures were measured after 24 and 72 h of incubation. To measure the pigment contents, the cultures were centrifuged for 10 min at 8000 rpm, and the supernatants were discarded. The chlorophyll *a*(Chl *a*) and carotenoids were extracted from cell pellets using 80% acetone for 24 h in darkness at 4 °C. Absorbance of the extracts was measured using a spectrophotometer at 663, 645, and 450 nm. The following equations set by Richards et al. (1952) [25] were used for the calculation of the pigment contents.

### 2.2. Measurement of Chlorophyll Fluorescence Transients

The polyphasic rise of chlorophyll fluorescence transients was detected by PEA (AquaPen-C AP-C 100, Photon Systems Instruments, The Czech Republic). The JIP transients were measured after exposure to [Hmim]Cl stress for 24 h and 72 h, respectively. Cells from the control and treated cultures were placed on special plastic clips. The parameters of absorption (ABS/RC), trapping (TR_0_/RC), electron transport (ET_0_/RC), dissipation (DI_0_/RC), the probability that a trapped exciton moves an electron into the electron transport chain beyond QA (*ψ*_0_), the maximum quantum yield of primary photochemistry (*φ*P_0_), the quantum yield of electron transport (*φ*E_0_), the probability that an absorbed photon is dissipated (*φ*D_0_), relative fluorescence values [(1 − *V_K_*/*V_J_*)_r_], and the performance indexes of photosystem II were calculated from the chlorophyll fluorescence transients [26,27]. The intensity of actinic light was set at 30 μmol photons/(m^2^·s). The equations used for calculating the above parameters were as follows [27]:ABS/RC = (*M*_0_*/V_J_*)(1 − *F*_0_/*F_M_*)(1)
TR_0_/RC = *M*_0_/*V_J_*(2)
ET_0_/RC = (TR_0_/RC)*ψ*_0_(3)
DI_0_/RC = (ABS/RC) − (TR_0_/RC)(4)
*φ*P_0_ = 1 − (*F*_0_/*F_M_*)(5)
*ψ*_0_ = 1 − *V_J_*(6)
*φ*E_0_ = (1 − *F*_0_/*F_M_*)(1 − *V_J_*)(7)
*φ*D_0_ = *F*_0_/*F_M_*(8)
PI_ABS_ = [RC/ABS][*φ*P_0_/(1 − *φ*P_0_)][*ψ*_0_/(1 − *ψ*_0_)](9)
(1 − *V_K_*/*V_J_*)_r_ = [(1 − *V_K_*/*V_J_*)_treatment_]/[(1 − *V_K_*/*V_J_*)_Control_].(10)

### 2.3. Observation of Cell Ultrastructure

Changes in the ultrastructure of the treated *M. aeruginosa* cells at 10 and 30 mg L^−1^ of [Hmim]Cl were detected using transmission electron microscopy. After 48 h of exposure, culture cells were centrifuged for 15 min at 4000 rpm under 4 °C, and the supernatant was discarded. The cell pellets were fixed with 2.5% glutaraldehyde solution. The samples were sent to the Institute of Hydrobiology, Chinese Academy of Sciences, for electron microscopic analysis. Changes in the ultrastructure of the treated cells were identified by comparison with the control group.

### 2.4. Transcription of mcyB Gene

#### 2.4.1. RNA Extraction and Reverse Transcription

The cells of *M. aeruginosa* PCC 7806 were harvested after 24 h and 72 h of exposure to each concentration of [Hmim]Cl. The cultures were centrifuged at 6000 rpm for 10 min, and the supernatant was discarded. Cell pellets were re-suspended in a Trizol reagent (Invitrogen, Waltham, MA, USA) and then frozen and thawed for three times. Total RNAs were extracted following the manufacturer’s manual of Trizol reagent and dissolved with diethyl pyrocarbonate-treated water. Reverse transcription was conducted using a Transcriptor First Strand cDNA Synthesis Kit (Roche, Germany).

#### 2.4.2. Determination of *mcyB* Gene Transcription

The primers 30F and 108R were used for the detection of the *mcyB* gene [16], whereas 5′-GGACGGGTGAGTAACGCGTA-3′ and 5′-CCCATTGCGGAAAATTCCCC-3′ were used to amplify 16S rRNA gene as the internal control [28]. The transcription of *mcyB* in *M. aeruginosa* was determined by qPCR after 24 h and 72 h of exposure to each concentration of [Hmim]Cl. The qPCR reaction was prepared with final volumes of 10 μL containing 5 μL of Master Mix (SYBR Green, CWBIO, China), 0.2 μL of each primer (10 μmol L^−1^), and 0.4 μL cDNA and 4 μL of sterile water. The qPCR conditions were set as follows: 95 °C for 10 min; 40 cycles at 95 °C for 15 s, 59 °C for 30 s, and 72 °C for 30 s. The relative transcription level of the *mcyB* gene was calculated using 2^−ΔΔCt^ where ΔΔCt = (Ct _target gene_ − Ct _16S rrn_)_stress_ − (Ct _target gene_ − Ct _16Srrn_)_control_ [29].

### 2.5. Statistical Analysis

The experimental data were analyzed by SPSS 20.0 (ANOVA, LSD). Differences were considered to be significant at *p* < 0.05.

## 3. Results

### 3.1. Inhibition of [Hmim]Cl on M. aeruginosa PCC 7806 Growth

As displayed in Figure 1a, cell growth was not affected by [Hmim]Cl at the concentrations of 5 and 10 mg L^−1^ at 24 h. However, cell growth was significantly inhibited by higher concentrations of [Hmim]Cl with 25% decrease at 30 mg L^−1^. After 72 h, the cell density of the control culture and the culture exposed to 5 mg L^−1^ of [Hmim]Cl increased by 118% and 61%, respectively. For cultures exposed to higher concentrations of [Hmim]Cl, cell density decreased after 72 h of exposure. No growth was observed at 10 mg L^−1^ of [Hmim]Cl, and cell density decreased by 23% in culture exposed to 30 mg L^−1^ of [Hmim]Cl at 72 h compared with 0 time. [Hmim]Cl at the concentrations of 10, 20, and 30 mg L^−1^ inhibited cell growth by >50% when compared with the control culture after 72 h of incubation. Regression curves of the logarithmic values of [Hmim]Cl concentrations against the inhibitory rate after 72 h of exposure is shown in Figure 1b. The calculated EC_50_ of [Hmim]Cl on the growth of *M. aeruginosa* was 10.624 ± 0.221 mg L^−1^.

### 3.2. Effects of [Hmim]Cl on the Pigment Contents

The effects of [Hmim]Cl on the pigment contents were similar to those on the cell growth (Figure 2). The Chl *a* concentration in the control culture increased by 34.5% and 174.2% after 24 h and 72 h of incubation, respectively. However, little changes were observed in the treated cultures at 24 h when compared with 0 time. Although the Chl *a* concentrations in the treated cultures increased after 72 h of exposure, they remained significantly lower than in the control culture. The inhibition of [Hmim]Cl on carotenoids production was more apparent than that on Chl *a* production. The increase of the carotenoids’ concentrations in the treated cultures was lower than 50% in comparison to three times in the control culture after 72 h of incubation.

### 3.3. Characteristics of Chlorophyll Fluorescence Transients

The parameters of the JIP test are illustrated as a radar plot (Figure 3). At 24 h, TR_0_/RC, *φ*P_0_, *φ*D_0_, and (1 − *V_K_*/*V_J_*)_r_ were not affected by [Hmim]Cl, whereas PI_ABS_ was significantly inhibited at all concentrations of [Hmim]Cl. The parameters ABS/RC and DI_0_/RC at 5 mg L^−1^ were significantly larger than those of the control culture and ET_0_/RC and *ψ*_0_ were significantly lower at 30 mg L^−1^.

The effects of [Hmim]Cl on chlorophyll fluorescence transients were more apparent at 72 h compared with those at 24 h. The φP_0_ value was significantly lower than that of the control culture after 72 h of exposure. DI_0_/RC was doubled at 5 and 20 mg L^−1^ and increased 3 times at 10 mg L^−1^ when compared with the values at 24 h. The ABS/RC was significantly increased at all concentrations at 72 h, whereas PI_ABS_ was significantly inhibited.

### 3.4. Effect of [Hmim]Cl on the Cell Ultrastructure

Figure 4 shows the cell ultrastructure of *M. aeruginosa* PCC 7806 at 10 and 30 mg L^−1^ of [Hmim]Cl after 48 h of incubation. Exposure to 10 mg L^−1^ of [Hmim]Cl resulted in cell damage including thylakoid disorganization and nucleoid diffusion. More severe cell damage was observed at 30 mg L^−1^ of [Hmim]Cl, including extensive disassembly of intracellular structures and disorganization of the cell wall and cell membrane.

### 3.5. Transcription of mcyB Gene

The relative expression of *mycB* gene at different concentrations of [Hmim]Cl was significantly lower than that of the control culture (Figure 5). With the exception of 5 mg L^−1^, the gene expressions in treated cultures were increased significantly after 72 h of incubation.

## 4. Discussion

Recently, several studies have been published regarding the structural characteristics, toxicity, environmental fate, and application of ILs [30,31]. Although ILs have been validated as green solvents instead of the hazardous organic solvents, they can negatively affect the aquatic environment [3].

A toxicological assessment of ILs in aquatic organisms involves bacteria (*Vibrio fischeri*), green algae (*Selenastrum capricornutum*, *Oocystis submarina*), diatoms (*Cyclotella meneghiniana*), mollusks (*Physa acuta*), vertebrates (*Danio rerio*, *Rana nigromaculata*, *Cyprinus carpio*), and plants (*Lemna minor*) [7]. A few kinds of aquatic organisms are inferior and simple, and their cell membrane is the first target that can be affected by ILs [32,33]. ILs could cause the destruction of a membrane lipid structure and the change of composition due to their properties as ionic surfactants. The long alkyl chain of [Hmim]Cl in the cationic part increases the lipophilicity and consequently interacts with the phospholipid bilayer and hydrophobic part of membrane proteins. This leads to the leakage of cellular contents and then cell death [34]. Such interpretation can explain the decrease of the *M. aeruginosa* cell density in the current study. The experimental results revealed that the EC_50_ (72 h) of [Hmim]Cl for the growth of *M. aeruginosa* PCC 7806 was 10.624 ± 0.221 mg L^−1^, referring to the low biodegradability as mentioned in the previous studies on imidazole or pyridine ILs [35,36,37].

Previous study revealed that ILs can inhibit algal photosynthesis by causing a decrease in the content of chlorophyll, thereby affecting the growth of algae [38,39]. Similarly, the contents of both Chlorophyll *a* and carotenoids were significantly decreased after 72 h of exposure to [Hmim]Cl. These results may be attributed to the long alkyl chain of [Hmim]Cl, which can lead to a leakage of cellular contents [30,37,40].

Chlorophyll fluorescence transients could be used to indicate the structure and function of PSII [27] and have become an important method to examine the response of plants and cyanobacteria to environmental stress [41]. According to the energy-flow model diagram of Strasser and Strasser (1995) [20], *φ*P_0_ can represent the electron-transfer performance of the electron donor side of the PSII reaction center. After 72 h of exposure, *φ*P_0_ was significantly inhibited, indicating that the electron donor side of the PSII reaction center of *M. aeruginosa* is the target site of [Hmim]Cl. *ψ*_0_ and *φ*E_0_ can reflect the performance of the electron acceptor side of the PSII reaction center. Inhibition of *ψ*_0_ and *φ*E_0_ by [Hmim]Cl indicated that the electron acceptor side of the PSII reaction center is also the target site of [Hmim]Cl. ABS/RC, TR_0_/RC, DI_0_/RC, and *φ*D_0_ values increased after extending the exposure time. These results indicate that although exposure to [Hmim]Cl leads to partial reaction center inactivation or cleavage, the efficiency of the active reaction center is improved. The values of (1 − *V_K_*/*V_J_*)_r_ can represent the state of the oxygen-evolving complex of PS II [26]. However, the relative value of (1 − *V_K_*/*V_J_*)_r_ showed no significant changes when compared with the control culture (Figure 3). Therefore, the oxygen-evolving complex was not the target of [Hmim]Cl. On the other hand, PI_ABS_, which is the performance index of photosynthesis, was markedly decreased in all treatments. The results from this study show that [Hmim]Cl would cause serious damage to the photosynthetic system of *M. aeruginosa*. It also proves that PI_ABS_ is more sensitive to reflect the changes of the photosynthetic system.

Previous investigation reported that ionic liquids can impair cells via lipid peroxidation [42] and DNA damage [43]. The results of ultrastructural observation by TEM indicate that both the thylakoid membrane and the cellular membrane were all damaged by [Hmim]Cl at 30 mg L^−1^. These results suggest that the mechanism of the toxicity of [Hmim]Cl against *M. aeruginosa* PCC 7806 may also be involved in the damage of membrane. The experimental data show that [Hmim]Cl had a short-time strong inhibition on the transcription of *mcyB*, which recovered slowly after 72 h. Given that there is a positive relationship between the growth rate and microcystin production [44], the rapid inhibition of *mcyB* transcription provided further evidence for the toxic effects of [Hmim]Cl on *M. aeruginosa* at transcription level. It also reflects that the potential threat indicated by [Hmim]Cl caused the cells to allocate energy toward cell division instead of toxin production.

The results of this study found that an adverse effect was showed on *M. aeruginosa* under different concentrations of [Hmim]Cl, but this is just the indoor research. If many ILs flowed into the aquatic ecosystem, threats may be superimposed on the algae with other substances, such as heavy metals and pesticides. The combined toxicity of ILs requires further study.

## 5. Conclusions

The main conclusions may be summarized as follows.

i.The EC_50_ of [Hmim]Cl on *M. aeruginosa* PCC 7806 was 10.624 ± 0.221 mg L^−1^ after 72 h of exposure.ii.[Hmim]Cl could destruct the electron-accepting side of the photosystem II of M. aeruginosa PCC 7806.iii.Cellular ultrastructure examination indicated that the distortion of the thylakoid membrane and the loss of the integrity of the cell membrane were associated with [Hmim]Cl treatment and concentration.iv.The transcriptional profiles of *mcyB* were depressed.

## Figures and Tables

**Figure 1 ijerph-19-08719-f001:**
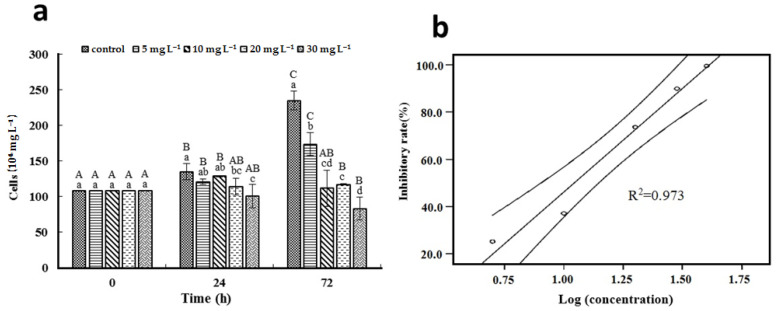
Effect of [Hmim]Cl on *M. aeruginosa* PCC 7806 growth. (**a**) Cell density after 24 h and 72 h of incubation. (**b**) Regression analysis between [Hmim]Cl concentration and inhibitory rate after 72 h. Average values ± standard deviation (*n* = 3). Different uppercase letters represent significant differences (*p* < 0.05) between exposure times within the same treatment. Different lowercase letters represent significant differences (*p* < 0.05) between treatments within the same exposure time.

**Figure 2 ijerph-19-08719-f002:**
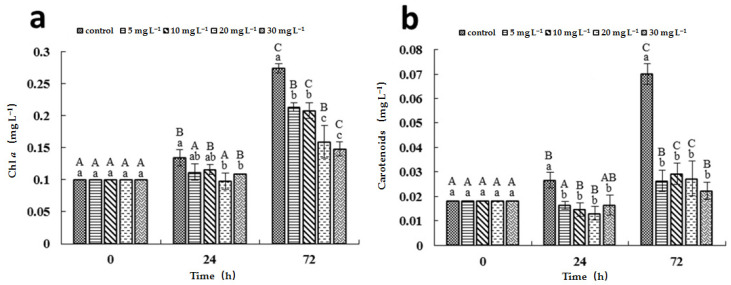
Effects of [Hmim]Cl on the photosynthetic pigment content of *M. aeruginosa* PCC 7806. (**a**) Chl *a* concentration; (**b**) carotenoids concentration. Different uppercase letters represent significant differences (*p* < 0.05) between exposure times within the same treatment. Different lowercase letters represent significant differences (*p* < 0.05) between treatments within the same exposure time.

**Figure 3 ijerph-19-08719-f003:**
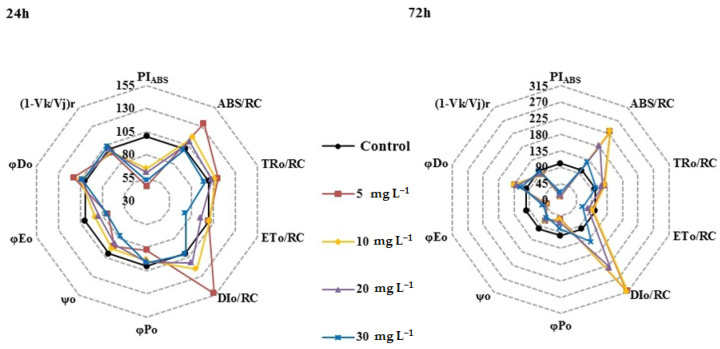
Radar plots of the parameters originated from chlorophyll fluorescence transients after 24 h and 72 h of [Hmim]Cl treatment.

**Figure 4 ijerph-19-08719-f004:**
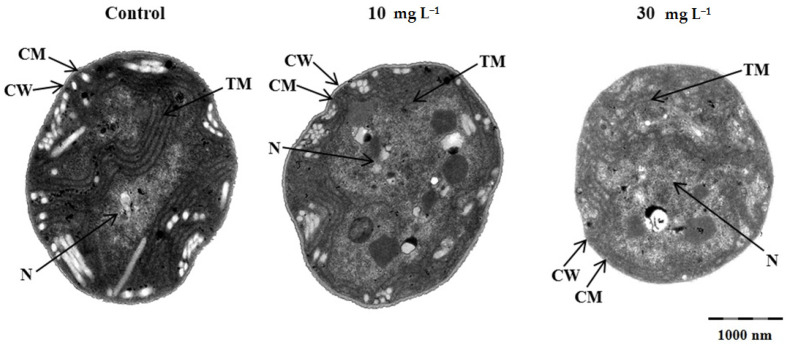
Effects of [Hmim]Cl on the ultrastructure of *M. aeruginosa* PCC 7806 after 48 h of exposure at 10 and 30 mg L^−1^, CW: cell wall; CM: cell membrane; TM: thylakoid membrane; N: nucleoid.

**Figure 5 ijerph-19-08719-f005:**
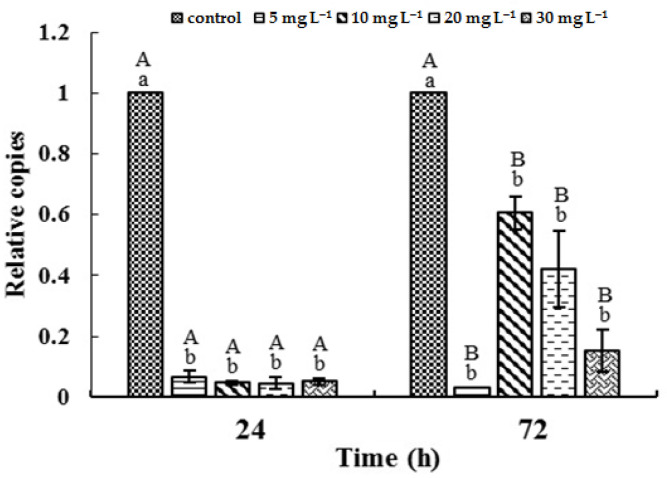
Effects of increasing [Hmim]Cl concentrations on the transcription of *mcyB* gene. Different uppercase letters represent significant differences (*p* < 0.05) between exposure times within the same treatment. Different lowercase letters represent significant differences (*p* < 0.05) between treatments within the same exposure time.

## Data Availability

The authors declare that all data and materials are available to be shared on a formal request.

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
