# Peer review of "Inhibition Effect of Ionic Liquid [Hmim]Cl on Microcystis Growth and Toxin Production"

_ijerph, 2022, doi:10.3390/ijerph19148719_

Round 1

Reviewer 1 Report

Although the work “Inhibition of ionic liquid [Hmim]Cl on Microcystis growth and toxin production” is promising as progression in the toxicity world. However, English has to be improved for a more fluent and understandable reading experience. Furthermore, sentences out of context are used to block or interrupt speech at various points throughout the manuscript, sometimes making communication difficult. As a result, the entire text should be evaluated to improve the English expression and eliminate or amend those aspects that obstruct the discourse's common thread. you may need to have your manuscript checked by a native English-speaking colleague or use a professional English editing service. Discussion section is poorly written. Needs to be improved. Conclusion should be based on the results. Overall, this research is quite promising. Therefore, I recommended it for minor revision.

Author Response

Although the work “Inhibition of ionic liquid [Hmim]Cl on Microcystis growth and toxin production” is promising as progression in the toxicity world. However, English has to be improved for a more fluent and understandable reading experience. Furthermore, sentences out of context are used to block or interrupt speech at various points throughout the manuscript, sometimes making communication difficult. As a result, the entire text should be evaluated to improve the English expression and eliminate or amend those aspects that obstruct the discourse's common thread. you may need to have your manuscript checked by a native English-speaking colleague or use a professional English editing service. Discussion section is poorly written. Needs to be improved. Conclusion should be based on the results. Overall, this research is quite promising. Therefore, I recommended it for minor revision.

Response : Thanks for this valuable advice. Thank you again for your recognition and support of this research work. We have revised the language of the manuscript, and sent it to a professional language polishing company for modification. The discussion section has been revised and new literature has been added. The conclusion part is rewritten. I hope the revised article which could get your approval. Thank you again for your review.

Reviewer 2 Report

Dear authors,

I consider that your work is valuable, it is a complete study that reveals the consequences of the use of ILs as "green solvents", in essential  components of the trophic network such as the primary producers. I suggest adding the graphical abstract and addressing some clarifications that you will find within the document.

Best regards

Author Response

Point 1:  Please allign the number of equation

Response 1: Thanks for this valuable advice and we have revised it.

Point 2:  Could you correct the inoculum concentration where ppropiated, since in the methodology section it is indicate that 1x106 was used and in the graphs 1a, 2a 1x104.

Response 2: Thank you for your careful revision, and we have have revised it.

Point 3:  The images are available at 72 hours?, since it would be approppiated to present the damage experience by the cells in the time reported in the analysis of the cell growth, pigment and chlorophyll content. 

Response 3: Thank you for your valuable question. According to the existing experiments, we selected an intermediate concentration and a high concentration group for observation. We want to observe how Microcystis cells behave under high concentrations of stress. It's consistent with our hypothesis and with existing research.

Point 4:  Maybe you can include this study

https://www.mdpi.com/1422-0067/21/17/6267

Response 4: Thank you for your advice. This is a good overview of ILs. We finished this manuscript early, so we didn't find this article then. And the references have been added.

Reviewer 3 Report

Dear Editor and Authors,

I have got the opportunity to write a review of the paper titled: "Inhibition of ionic liquid [Hmim]Cl on Microcystis growth and toxin production”, written by seven authors: Yang Liu, Yi Jie Zhang, Yousef Sultan, Peng Xiao, Li Yang, Han Yang Lu, Bang Jun Zhang, from China and Egypt

After carefully checking, I can give only a negative recommendation of this paper for publication in MDPI - International Journal of Environmental Research and Public Health. Below are some detailed instructions that can be useful for corrections and resubmissions to other journals. The taxonomic mess significantly discourages the reader. My primary concern is that, in these modern days, the authors should include a more completed investigation in terms of the profile of other important artificial inhibitors related to physiological performance at the photosynthesis level and could help solve the bottlenecks for toxins production by blue-green species. My second caveat is that the results of these studies do not have any practical application in natural and semi-natural aquatic ecosystems plagued by cyanobacterial blooms.

Title. A more concise title will get the reader interested faster.

Abstract. I would include a bit more information about failures in the results. However, the abstract length should not be significantly longer. As the article deals with important issues, the abstract should be less related to its conclusions.

Where is the abbreviation list?

The aim(s) of the work is/are unclear; no hypothesis or assumption? The main research objectives are missing in the article (are unclear), especially in the introduction, where it has to be written.

At the end of INTRO, is it a part of the methodology?  “Growth rate, photosynthetic pigment content and chloro-phyll fluorescence transient parameters of M. aeruginosa were measured under [Hmim]Cl stress. The effects of [Hmim]Cl on the cell structure of M. aeruginosa were detected by transmission electron microscopy, and the expression of microcystin gene was examined by quantitative PCR (qPCR).

The influence of 1-hexyl-3-methylimidazolium chloride on native water organisms (phytoplankton, zooplankton, fish) is missed.

Add the author(s) of the taxa description – full name of microalgae species (with citation and only on the first mention) of Microcystis aeruginosa (Kützing) Kützing 1846, please see AlgaBase.

Why is there no photographic documentation of the Microcystis aeruginosa strain?

The methods are suitable but not supported by figures or photos from the analysis time.

Deficient quality of figures 1-4. The graphs are unreadable. The TEM photos are too small. Now it isn't easy to orientate yourself.

The number of repetitions is too small, in my opinion. The data obtained are not statistically significant.

The discussion chapter is incomplete, e.g. the influence of environmental conditions (pH, salinity, temperature, and others…) on the aspects of growth, distribution patterns on Earth and toxin production (in case of blooms) by Microcystis spp. are neglected.

The content of the conclusions looks such as the abstract. I suggest completing the “conclusions”, which should positively influence the whole article.

I hope that these amendment proposals will be passed in the edition process.

Best regards,

Reviewer

Author Response

Response to Reviewer 3 Comments

Point 1:  A more concise title will get the reader interested faster.

Response 1: Thanks for this valuable advice. The first one we used was “Effect of 1-hexyl-3-methylimidazolium chloride on growth, photosynthesis of toxigenic cyanobacterium Microcystis aeruginosa 7806 and gene expression of microcystins production”, and now we changed it to “Inhibition of ionic liquid [Hmim]Cl on Microcystis growth and toxin production”.

Point 2:  I would include a bit more information about failures in the results. However, the abstract length should not be significantly longer. As the article deals with important issues, the abstract should be less related to its conclusions.

Response 2: Thanks for this valuable advice, and we have revised it as follows,

Ionic liquids (ILs) are known as “green solvents”, and widely used in industrial applications. However, little research has been conducted on cyanobacteria This study was conducted to investigate the toxicity of ionic liquids ([Hmim]Cl) on Microcystis aeruginosa PCC7806. The EC50 (72 h) of [Hmim]Cl on the growth of M. aeruginosa PCC 7806 was 10.624 ± 0.221 mg L-1. The possible mechanism of toxicity of [Hmim]Cl against M. aeruginosa PCC 7806 was evaluated by measuring cell growth, photosynthetic pigment contents, chlorophyll fluorescence transients, cell ultrastructure, and transcription of the microcystin-producing gene (mcyB). The contents of chlorophyll a and carotenoids were significantly decreased in treated M. aeruginosa cultures. The results of chlorophyll fluorescence transients showed that [Hmim]Cl could destruct the electron accepting side of the photosystem II of M. aeruginosa PCC 7806. Transmission electron microscopy demonstrated cell damage including changes in the structure of the cell wall and cell membrane, thylakoid destruction, and nucleoid disassembly. Transcription of the mcyB gene was also inhibited under [Hmim]Cl stress. In summary, this study provided new insights into the toxicity of [Hmim]Cl on cyanobactreia.

Point 3:  The aim(s) of the work is/are unclear; no hypothesis or assumption? The main research objectives are missing in the article (are unclear), especially in the introduction, where it has to be written. 

Response 3: Thanks for this valuable advice, and we have revised it as follows,

The 1-hexyl-3-methylimidazolium chloride, which is abbreviated as [Hmim]Cl, is one of the widely used ILs. Up to date, there is no available study concerning the effects of [Hmim]Cl on cyanobacteria. The current study aimed to illustrate the possible toxicological mechanism of [Hmim]Cl on cyanobacteria from the view of cell growth, photosynthesis and gene expression.

Point 4:  At the end of INTRO, is it a part of the methodology?  “Growth rate, photosynthetic pigment content and chloro-phyll fluorescence transient parameters of M. aeruginosa were measured under [Hmim]Cl stress. The effects of [Hmim]Cl on the cell structure of M. aeruginosa were detected by transmission electron microscopy, and the expression of microcystin gene was examined by quantitative PCR (qPCR).

Response 4: Thanks for this valuable advice, and we have deleted the redundancy contents.

Point 5:  The influence of 1-hexyl-3-methylimidazolium chloride on native water organisms (phytoplankton, zooplankton, fish) is missed. 

Response 5: Thanks for this valuable advice and we have added it in our discussion.

Point 6: Add the author(s) of the taxa description – full name of microalgae species (with citation and only on the first mention) of Microcystis aeruginosa (Kützing) Kützing 1846, please see AlgaBase.

Response 6: Thank you for your careful revision, and we have revised it.

Point 7: Why is there no photographic documentation of the Microcystis aeruginosa strain?

Response 7: Thank you for your advice. Since this study is mainly concerned with toxicological properties, we do not show pictures of Microcystis aeruginosa, and we can provid it as supplementary materials.

Point 8: Deficient quality of figures 1-4. The graphs are unreadable. The TEM photos are too small. Now it isn't easy to orientate yourself.

Response 8: Thank you for your advice. Since the figure in the article is a screenshot, we will provide it as an attachment.

Point 9: The number of repetitions is too small, in my opinion. The data obtained are not statistically significant.

Response 9: Thank you for your question. A pre-experiment was conducted in this study, so we selected the relevant repetition times based on the pre-experiment. Thank you for your rigorous scientific attitude.

Point 10: The discussion chapter is incomplete, e.g. the influence of environmental conditions (pH, salinity, temperature, and others…) on the aspects of growth, distribution patterns on Earth and toxin production (in case of blooms) by Microcystis spp. are neglected.

The content of the conclusions looks such as the abstract. I suggest completing the “conclusions”, which should positively influence the whole article.

Response 10: Thank you for your advice, and we have revised this section.

Reviewer 4 Report

The authors studied the possibility of using ionic liquids (ILs), namely [Hmim]Cl, as green organic solvents in synthetic and analytical processes instead of conventional volatile organic compounds. For this purpose, they evaluated the potential effects of ILs’ use on aquatic ecosystems by selecting a cyanobacterial aquatic species as study model.

The impact of ILs on the species Microcystis aeruginosa was studied by measuring different parameters as cell growth, photosynthesis and gene expression.

The introduction needs to be improved, because references to recent relevant papers on the environmental risks connected with the ILs use, such as for Khan et al.2016 and 2021 (https://doi.org/10.1016/j.proeng.2016.06.567, https://doi.org/10.1016/j.jhazmat.2021.125364) and Flieger and Flieger 2020 (https://www.ncbi.nlm.nih.gov/pmc/articles/PMC7504185/), are missing. These could be eventually useful in discussion paragraph, as well as papers by Hu et al 2021 (https://doi.org/10.1016/j.ecoenv.2020.111629), and Kumari et al., 2020 (https://doi.org/10.1007/s12551-020-00754-w)

Conclusions should not be in the form of an abstract. Here I expect to find responses, even if only hypotheses, to the potential use of ILs as organic solvents….

Some minor additional comments and suggestions are in below:

Introduction

Row 8

I suggest to change “they may pose ecological risk on aquatic ecosystem with high solubility in water “ with “their high solubility in water may pose an ecological risk to aquatic ecosystems.”

Row 13

I suggest to change “due to quickly reaction to any environmental changes “ with “because of their rapid response to environmental changes”

Page 2 row 6

has instead of have

Page 2 row 7

Is instead of are

Page 2 Row 19-20

Please provide bibliographic references

Material and Methods

Paragraph 2.2, row 1

Insert a space after Cl, and replace were with was

Row 2 insert a space after of

Paragraph 2.3

Row1

Replace in with of, and was detected instead of were detected

Rows 10-19

Are the numbers reported on the right side the relative references? References should be added

Paragraph 2.4 Row 1

Specify why you chose 10 and 30 mg L-1 concentrations after 48h to monitor ultrastructural changes

Paragraph 2.5.1 Row 1 and Paragraph 2.5.2 Row 5

Please add ‘concentrations’ after [Hmim]Cl

Paragraph 2.5.2 Row 8

At 95°C instead of of 95°C

Page 3

Caption of Figure 1.

M. aeruginosa in italics

Results

Caption of Figure 3

Treatments instead of treatment

Caption of Figure 4

M. aeruginosa in italics

Replace the comma with a foolstop before CW

Paragraph 3.2 Row  5

I suggest to replace ‘they were significantly lower than that of the control culture’ with ‘they remained significantly lower than in the control culture’

Paragraph 3.5 Row 3

Replace at with in

Caption of figure 5

Line 1

‘Effects of increasing  [Hmim]Cl concentrations on the…’ instead of ‘Effects of [Hmim]Cl on the…’

Discussion

Row 5

Replace part with target

 The growth of M. aeruginosa was significantly inhibited by [Hmim]Cl with an EC50 (72 h) of 10.624 ± 0.221 mg L-1.

photosynthetic pigment contents, with inhibition effects by carotenoids and chloprophyll (photosystem II was a probable target)   

(i)              cell ultrastructure with several morphological damages, and

(ii)            inhibition in the transcription of the mcyB gene

Author Response

Point 1:  Row 8

I suggest to change “they may pose ecological risk on aquatic ecosystem with high solubility in water “ with “their high solubility in water may pose an ecological risk to aquatic ecosystems.”

Response 1: Thanks for this valuable advice and we have replaced this sentence. (in red)

Point 2:  Row 13 

I suggest to change “due to quickly reaction to any environmental changes “ with “because of their rapid response to environmental changes”

Response 2: Thank you for your careful revision, and we have replaced this sentence. (in red)

Point 3:  Page 2 row 6    has instead of have 

Response 3: Thank you for your careful revision, and we have revised it. (in red)

Point 4:  Page 2 row 7    Is instead of are

Response 4: Thank you for your careful revision, and we have corrected the word. (in red)

Point 5:  Page 2 Row 19-20   Please provide bibliographic references 

Response 5: Thanks for this valuable advice and we have added the references. (in red)

Point 6: Paragraph 2.2, row 1  Insert a space after Cl, and replace were with was. Row 2 insert a space after of

Response 6: Thank you for your careful revision, and we have revised it. (in red)

Point 7: Paragraph 2.3  Row1  Replace in with of, and was detected instead of were detected

Response 7: Thank you for your careful revision, and we have revised this sentence. (in red)

Point 8: Rows 10-19  Are the numbers reported on the right side the relative references? References should be added

Response 8: Thank you for your careful revision, and we have added the references. (in red)

Point 9: Paragraph 2.4 Row 1  Specify why you chose 10 and 30 mg L-1 concentrations after 48h to monitor ultrastructural changes

Response 9: Thank you for your question. According to the existing experiments, we selected an intermediate concentration and a high concentration group for observation.

Point 10: Paragraph 2.5.1 Row 1 and Paragraph 2.5.2 Row 5  Please add ‘concentrations’ after [Hmim]Cl

Response 10: Thank you for your good advice, and we have added ‘concentrations’ after [Hmim]Cl. (in red)

Point 11: Paragraph 2.5.2 Row 8   At 95°C instead of of 95°C

Response 11: Thank you for your careful revision, and we have revised it. (in red)

Point 12: Caption of Figure 1. M. aeruginosa in italics

Response 12: Thank you for your careful revision, and we have revised it. (in red)

Point 13: Caption of Figure 3 Treatments instead of treatment

Response 13: Thank you for your careful revision, and we have revised it. (in red)

Point 14: Caption of Figure 4  M. aeruginosa in italics  Replace the comma with a foolstop before CW

Response 14: Thank you for your careful revision, and we have revised it. (in red)

Point 15: Paragraph 3.2 Row  5

I suggest to replace ‘they were significantly lower than that of the control culture’ with ‘they remained significantly lower than in the control culture’

Response 15: Thank you for your good advice, and we have replaced this sentence. (in red)

Point 16: Paragraph 3.5 Row 3  Replace at with in

Response 16: Thank you for your careful revision, and we have revised it. (in red)

Point 17: Caption of figure 5 Line 1

‘Effects of increasing  [Hmim]Cl concentrations on the…’ instead of ‘Effects of [Hmim]Cl on the…’

Response 17: Thank you for your careful revision, and we have replaced the words. (in red)

Point 18: Discussion  Row 5  Replace part with target

Response 18: Thank you for your careful revision, and we have revised it. (in red)

Round 2

Reviewer 4 Report

Dear authors, 

thank you for considering my suggestions. I only have a few more remarks:

1) add to the bibliography the reference Hu et al., 2021 cited at the last row of pag 6

2) Begin paragraph 2 with 'Microcystis' rather than the letter 'M.'

3) Write M. aeruginosa not to italics In the title of paragraph 3.1 

4) Write in italics all the names of taxa included in the discussion paragraph (rows 6-9)

5) I propose beginning the conclusion paragraph with the sentence: 'Main conclusions may be summarized as follows:' 

Author Response

Thanks a lot for your precious comments concerning our manuscript. We have carefully considered all the comments and revised the manuscript accordingly. The responses to the comments are presented in red following.

Point 1:  add to the bibliography the reference Hu et al., 2021 cited at the last row of pag 6

Response 1: Thanks for this valuable advice and we have revised it.

Point 2:  Begin paragraph 2 with 'Microcystis' rather than the letter 'M.'

Response 2: Thank you for your careful revision, and we have revised it.

Point 3:  Write M. aeruginosa not to italics In the title of paragraph 3.1

Response 3: Thank you for your careful revision, and we have revised it.

Point 4:  Write in italics all the names of taxa included in the discussion paragraph (rows 6-9)

Response 4: Very thanks for your careful revision, and we have revised all the names of taxa.

Point 5:  I propose beginning the conclusion paragraph with the sentence: 'Main conclusions may be summarized as follows:'

Response 5: Thanks for this valuable advice and we have added the sentence.
